# Persistently High Procalcitonin and C-Reactive Protein Are Good Predictors of Infection in Acute Necrotizing Pancreatitis: A Systematic Review and Meta-Analysis

**DOI:** 10.3390/ijms25021273

**Published:** 2024-01-20

**Authors:** Dorottya Tarján, Eszter Szalai, Mónika Lipp, Máté Verbói, Tamás Kói, Bálint Erőss, Brigitta Teutsch, Nándor Faluhelyi, Péter Hegyi, Alexandra Mikó

**Affiliations:** 1Centre for Translational Medicine, Semmelweis University, 1085 Budapest, Hungary; tarjan.dorottya@semmelweis.hu (D.T.); szalai.eszter@semmelweis.hu (E.S.); lipp.monika@semmelweis.hu (M.L.); koi.tamas@semmelweis.hu (T.K.); eross.balint@semmelweis.hu (B.E.); teutsch.brigitta@semmelweis.hu (B.T.); faluhelyi.nandor@pte.hu (N.F.); miko.alexandra@pte.hu (A.M.); 2Institute of Pancreatic Diseases, Semmelweis University, 1083 Budapest, Hungary; 3Institute for Translational Medicine, Medical School, University of Pécs, 7624 Pécs, Hungary; mate.verboi@stud.semmelweis.hu; 4Department of Restorative Dentistry and Endodontics, Semmelweis University, 1088 Budapest, Hungary; 5Department of Stochastics, Institute of Mathematics, Budapest University of Technology and Economics, 1111 Budapest, Hungary; 6Department of Radiology, Medical Imaging Centre, Semmelweis University, 1085 Budapest, Hungary; 7Division of Medical Imaging, Medical School, University of Pécs, 7624 Pécs, Hungary; 8Translational Pancreatology Research Group, Interdisciplinary Centre of Excellence for Research Development and Innovation, University of Szeged, 6725 Szeged, Hungary; 9Department for Medical Genetics, Medical School, University of Pécs, 7624 Pécs, Hungary

**Keywords:** scoring system, antibiotic therapy, sepsis, necrosis, infection

## Abstract

Infected necrotizing pancreatitis (INP) is associated with an increased risk of organ failure and mortality. Its early recognition and timely initiation of antibiotic therapy can save patients’ lives. We systematically searched three databases on 27 October 2022. In the eligible studies, the presence of infection in necrotizing pancreatitis was confirmed via a reference test, which involved either the identification of gas within the necrotic collection through computed tomography imaging or the examination of collected samples, which yielded positive results in Gram staining or culture. Laboratory biomarkers compared between sterile necrotizing pancreatitis and INP were used as the index test, and our outcome measures included sensitivity, specificity, the receiver operating characteristic (ROC) curve and area under the ROC curve (AUC). Within the first 72 hours (h) after admission, the AUC of C-reactive protein (CRP) was 0.69 (confidence interval (CI): 0.62–0.76), for procalcitonin (PCT), it was 0.69 (CI: 0.60–0.78), and for white blood cell count, it was 0.61 (CI: 0.47–0.75). After the first 72 h, the pooled AUC of CRP showed an elevated level of 0.88 (CI: 0.75–1.00), and for PCT, it was 0.86 (CI: 0.60–1.11). The predictive value of CRP and PCT for infection is poor within 72 h after hospital admission but seems good after the first 72 h. Based on these results, infection is likely in case of persistently high CRP and PCT, and antibiotic initiation may be recommended.

## 1. Introduction

Acute pancreatitis (AP) ranks among the most prevalent gastroenterological conditions, affecting many patients worldwide [1]. While interstitial edematous pancreatitis is the most common form, pancreatic necrosis occurs in approximately 5–10% of cases [2].

Pancreatic necrosis arises from compromised pancreatic perfusion, emphasizing the pivotal role of pancreatic ischemia and microcirculatory disturbances in acute pancreatitis development. This injury, whether primary or secondary to non-vascular causes, manifests early in acute pancreatitis and correlates with severity progression. Conversely, enhancing blood flow not only averts acute pancreatitis but also expedites recovery. Therefore, it is recommended to provide aggressive hydration to all patients, especially during the first 12–24 hours (h), unless restricted by cardiovascular or renal comorbidities [2,3,4].

AP is an inflammatory condition, thus exhibiting systemic manifestations of inflammation, including fever, tachycardia, hypotension, elevated white blood cell count (WBC), and increased levels of C-reactive protein (CRP) [5,6,7]. These characteristics do not differentiate between inflammation and infection, leading to an excessive utilization of antibiotics across the entire range of disease severity without distinguishing between the two [8].

Nonetheless, approximately 30% of individuals diagnosed with acute necrotizing pancreatitis (ANP) will experience debris infection due to the migration of intestinal microbial flora, resulting in infected necrotizing pancreatitis (INP). The presence of sepsis exacerbates the complexity of the condition, leading to alarmingly high mortality rates, as high as 40% [9,10].

Diagnosing and treating infected necrosis remains challenging. Antimicrobial therapy is most appropriate when there is a culture-proven infection in pancreatic necrosis or a strong suspicion of infection indicated by factors such as gas in the collection, bacteremia, sepsis, or clinical deterioration. Prophylactic antibiotics for preventing the infection of sterile necrosis are not recommended. Initiating enteral feeding early in patients with pancreatic necrosis is recommended to reduce the risk of infection. This preventive measure enhances the integrity of the mucosal barrier and reduces the likelihood of bacterial translocation in the gastrointestinal tract. The drainage and/or debridement of pancreatic necrosis is warranted in patients with infected necrosis. It is recommended to avoid pancreatic debridement in the early acute period (first 2 weeks) due to its association with increased morbidity and mortality; optimal debridement is ideally delayed for 4 weeks [11,12].

While several scoring systems have been developed to predict the severity of AP, there is currently a lack of scoring systems specifically designed to predict the presence of infection [13,14]. Identifying infection early is crucial for timely interventions and appropriate management, especially in high-risk patients who may benefit from early antibiotic therapy. To address this gap in the literature, we conducted a meta-analysis on laboratory markers to investigate their potential for predicting the presence of infection in AP. Our meta-analysis aimed to identify early predictors for INP.

## 2. Methods

### 2.1. Protocol

The meta-analysis was reported following the Preferred Reporting Items for Systematic Review (PRISMA) statement (Appendix A), and we followed the Cochrane Handbook [15,16]. The protocol was registered in the International Prospective Register of Systematic Reviews (PROSPERO) under registration number CRD42022370672.

### 2.2. Information Sources and Search Strategy

A systematic search was conducted in Medline (via Pubmed), Embase, and The Cochrane Central Register of Controlled Trials (CENTRAL) from inception to 27 October 2022 (Appendix A).

### 2.3. Eligibility Criteria

We used the PIRD (patient; index test; reference test; diagnosis of interest) framework to formulate our research question. We included randomized controlled trials, prospective and retrospective observational cohort studies with the following criteria: (1) the study was conducted in the adult population with necrotizing pancreatitis, and the diagnosis of AP adhered to the ‘two out of three’ criteria outlined in the International Association of Pancreatology and American Pancreatic Association guidelines, which included: (a) upper abdominal pain, (b) serum amylase or lipase levels elevated to at least three times the upper limit of normal, and (c) the presence of characteristic findings on pancreatic imaging; (2) the reference test confirmed infection by computed tomography imaging with the presence of gas in the necrotic collection or by examination of the sample acquired by an intervention using Gram staining or culture; (3) the index test was using any laboratory biomarker that was compared between patients suffering from sterile necrotizing pancreatitis (SNP) or INP; (4) and data on the sensitivity, specificity, receiver operating characteristic (ROC) curve or area under the ROC curve (AUC) to predict pancreatic infection, differentiating SNP from INP [2,17].

We excluded animal or in vitro studies, case reports, case series, and abstracts. There was no limitation on the publication language or date.

### 2.4. Selection Process

The publications were processed by the EndNote software (version 20.4.1.16297). After excluding duplicates, two reviewers (DT and ML) independently assessed the titles and abstracts. Later, the full text of relevant studies was obtained and independently evaluated. Cohen’s kappa was used to assess the level of agreement among the review authors. Disagreements between the two authors were resolved by a third author (AM). To identify additional eligible studies, we also investigated the references of relevant reviews or included articles. Also, CitationChaser was used for both backward and forward citation chasing of the included studies [18].

### 2.5. Data Collection Process and Data Items

The eligible articles were reviewed by two independent reviewers who extracted the following data using a pre-designed data table: first author, study location, study design, study duration, demographic information about the study population, number of cases of IPN, type of reference test, timing of laboratory test measurements, type of laboratory tests, and outcomes (laboratory biomarkers’ sensitivity, specificity, AUC, true positive, false positive, false negative, true negative, cut-off value).

### 2.6. Study Risk of Bias Assessment

Two authors (D.T. and M.L.) independently assessed the risk of bias in the included studies using the Quality Assessment of Diagnostic Accuracy Study-2 (QUADAS-2) Tool [19]. Each item was evaluated as having a low, unclear, or high risk of bias. Disagreements were resolved by a third reviewer (A.M.).

### 2.7. Certainty of Evidence

To assess the certainty of evidence, we employed the Grading of Recommendations Assessment, Development, and Evaluation (GRADE) methodology [20]. The GRADEpro tool was used to generate a table.

### 2.8. Synthesis Methods

Statistical analyses were carried out using the R statistical software (version 4.1.2.) and the R script of the online tool described by Freeman [21]. For all statistical analyses, a *p*-value of less than 0.05 was considered significant.

We collected the AUC values and the confidence intervals (CIs) of the different diagnostic scores from the studies. From the CIs, we estimated the standard deviations of the AUC values, and we applied classical inverse-variance random-effects meta-analysis with the restricted maximum likelihood estimator to gain pooled AUC estimates with 95% CI. A test’s discrimination ability is considered excellent when its AUC falls between 0.90 and 1.00, while AUC values in the ranges of 0.80 to 0.90, 0.70 to 0.80, 0.60 to 0.70, and 0.50 to 0.60 signify good, fair, poor, and failed discrimination, respectively [22]. As only a few studies contributed to the meta-analyses, the Hartung–Knapp adjustment was applied. The I^2^ measure, its confidence interval and the Cochrane Q test were used to examine heterogeneity. Heterogeneity levels were categorized as low, moderate, and high when I^2^ values were 25%, 50%, and 75%, respectively. We also performed subgroup analysis according to the time horizons. We did not assume that the standard deviations of the random effects were the same in the subgroups.

In the case of the short-term diagnostic performances of PCT and CRP, we collected from a sufficient number of studies the total number of patients with and without infected necrosis, sensitivity, and specificity values, in most cases, along with the corresponding thresholds. From these data, we calculated two-by-two contingency tables for each threshold containing the true positive, false positive, false negative, and true negative values. Since the thresholds differed across studies, we fitted the summary ROC (SROC) curve using the non-Bayesian version of the approach [23]. For clarity, we note that Harbor et al. show that the employed method is mathematically equivalent to the bivariate model [24,25,26]. We plotted on an ROC plot the resulting SROC curves and the study-level estimates with their confidence intervals. The SROC curve shows the trade-off between sensitivity and specificity as the employed threshold varies. Publication bias analysis was not possible, since the number of involved studies was less than 10.

## 3. Results

### 3.1. Search and Selection

Altogether, 7975 articles were identified through our systematic search strategy. After the duplicate removal, 5400 titles and abstracts were screened, leaving 122 studies for full-text review. We excluded 60 studies because they did not compare SNP and IPN. Most commonly, they included a comparison between severe AP and IPN. Another 48 articles were not eligible for the analysis because they did not contain any data for laboratory parameters (Appendix A). One study had to be excluded because of overlapping populations [27]. Using citation chasing, we discovered no new articles relevant to our research topic. Finally, eight [28,29,30,31,32,33,34,35] of the thirteen studies included in our analysis were deemed eligible for meta-analysis, while the remaining five [36,37,38,39,40] were considered for systematic review. The study selection process is shown in Figure 1.

### 3.2. Study Characteristics

All studies included in our analysis were observational studies with four being retrospective and nine being prospective in design (Table 1). The quantitative synthesis involved 758 patients diagnosed with necrotizing acute pancreatitis, of whom 324 were identified as having INP.

Nine articles were published before the 2012 Atlanta classification, in which peripancreatic necrosis was added as necrotizing pancreatitis [2].

INP was identified through computed tomography imaging with the presence of gas in the necrotic collection or examination of samples obtained during an intervention or fine-needle aspiration with Gram stain, culture, or both. Interventions were defined as percutaneous or endoscopic drainage or percutaneous, endoscopic, or surgical necrosectomy.

CRP and PCT were the most frequently described predictors. However, many laboratory parameters such as albumin, hematocrit (HCT), blood urea nitrogen (BUN), creatinine, lymphocyte count, lactate dehydrogenase (LDH), interleukin-6 (IL-6), tumor necrosis factor-alpha (TNF-alpha), soluble intercellular adhesion molecules (sICAM-1), soluble triggering receptor expressed on myeloid cells-1 (sTREM-1), presepsin, and granulocyte colony-stimulating factor (G-CSF) were investigated in one study only. Due to limited comparable studies, we were unable to conduct a quantitative analysis of these factors.

### 3.3. Within 72 h after Admission CRP, PCT, and WBC Levels Alone Have Poor Predictive Value in ANP

A subgroup analysis was performed based on the time of the index test. Timing of the index test was within the first 72 h after admission in five articles [28,29,30,31,36].

Our results confirmed that within the first 72 h after admission, the pooled AUC of CRP was 0.69 (CI: 0.62–0.76), for PCT, it was 0.69 (CI: 0.60–0.78), and for white blood cell count (WBC), it was 0.61 (CI: 0.47–0.75) (Figure 2, Figure 3 and Figure 4).

### 3.4. After the First 72 h of Admission CRP and PCT Levels Have Good Predictive Value in ANP

After the first 72 h of admission, in studies investigating a minimum two-week period of the disease, the pooled AUC of CRP showed an elevated level of 0.88 (CI: 0.75–1.00); for PCT, it was 0.86 (CI: 0.60–1.11), which shows that it had a good predictive value (Figure 2 and Figure 3) [22].

### 3.5. Risk of Bias Assessment

The QUADAS-2 scores indicate that the included studies were moderately high quality (Appendix A).

### 3.6. Certainty of Evidence

The certainty of evidence derived from our analysis was graded as low. This is primarily due to imprecision in the data, as evidenced by wide confidence intervals. The imprecision suggests a certain degree of uncertainty in the estimates, which could impact the overall certainty of the findings (Appendix A).

### 3.7. Publication Bias and Heterogeneity

Funnel plot analyses could not be appropriately performed due to the low number of studies. No significant heterogeneity was observed among the subgroups.

## 4. Discussion

Our meta-analysis found that the effectiveness of CRP and PCT in predicting infection in necrotizing pancreatitis can vary depending on the stage of the disease. Initially, within the first three days of admission, CRP, PCT, and WBC had low predictive accuracy for infection, aligning with previous findings by Párniczky et al. [8,22]. However, after the initial three days of admission, investigating at least two weeks of the period of the disease, CRP and PCT demonstrated good predictive accuracy for infection [23]. Repeated measurements and monitoring these biomarkers over time may be essential to achieve more reliable results.

PCT, the inactive 116-amino-acid propeptide of calcitonin, demonstrates significant elevation within 2 to 4 h in severe systemic inflammation or bacterial infections while showing a minimal increase in response to viral infections. In contrast, the elevation of CRP and the WBC count occurs gradually and reaches its maximum level approximately 36 h after exposure to endotoxin. CRP is an acute-phase protein and needs more than 72 h to reach its highest level. Previous publications have shown that it can predict the severity of acute pancreatitis, and pancreatic necrosis is strongly associated with a CRP level exceeding 150 mg/L within the initial 72 h period [5,41,42]. PCT production can be triggered by microbial antigens such as endotoxin and the immune response by stimulating cytokines like IL-1b, TNF-alfa, and IL-6 [43,44]. PCT temporarily increases for 12–24 h after surgery but returns to baseline within 48 h if no infection is present. CRP and WBCs may stay elevated longer post-surgery, regardless of infection [45]. The PROCAP randomized trial was conducted at a single center, where a PCT algorithm (threshold: 1 ng/mL) was employed to steer antibiotic therapy. The findings indicate a noteworthy decline in the likelihood of antibiotic prescription when employing the PCT algorithm. This reduction effectively curtailed the inappropriate usage of antibiotics without resulting in a substantial rise in infections or prolongation of hospitalization [46]. The trial results highlight the importance of utilizing biomarkers like PCT in clinical practice to optimize patient care and improve outcomes in INP. Generally, the threshold for antibiotic administration is “encouraged” for values exceeding 0.5 ng/mL but “strongly recommended” for values surpassing 1.0 ng/mL [43]. Another study identified the inability to decrease PCT levels to less than 60% of the baseline value after seven days of intervention as a prognostic indicator for mortality [47].

In a prior systematic review, PCT was the most accurate indicator of IPN [48]. Chen et al. conducted a study that demonstrated the potential of PCT as a reliable predictor of infection within the first 48 h. Their findings may be attributed to the fact that their tertiary center specializes in treating critically ill patients more prone to developing infections [29]. The included studies did not have a follow-up period, which might restrict the assessment of long-term outcomes or the detection of delayed complications. In the statistical analysis of CRP and PCT, pooling sensitivity and specificity was unattainable due to variations in threshold values. This was further compounded by the restriction imposed by the limited number of eligible studies, preventing their aggregation.

Systemic inflammation is thought to play a pivotal role in the pathophysiology of organ dysfunction, with cytokines acting as key mediators in regulating the inflammatory response. IL-6 is a proinflammatory cytokine produced by various cell types in response to tissue damage and stimuli such as TNF-alpha and IL-1β. IL-6 acts by inducing the production of acute-phase proteins in hepatocytes, including CRP. IL-6 demonstrates significantly elevated serum levels in cases of necrotizing pancreatitis and severe acute pancreatitis on the day of admission, making it an excellent marker for early severity stratification and predicting remote organ failure [31,49]. IL-6 levels were significantly elevated in SNP and IPN [38].

TNF-alpha and sICAM were explored as potential markers for IPN, but they did not demonstrate satisfactory predictive value [31,38]. One study examined the predictive value of G-CSF, building upon previous findings, suggesting a potential association between low G-CSF levels and increased infection risk. However, the study’s results revealed that the G-CSF concentration was slightly elevated in cases of INP and did not serve as a useful marker [32,50]. 

STREM-1 and presepsin (soluble CD14-ST) were investigated individually in separate studies, and the results showed AUC values of 0.792 and 0.956 [30,33]. In recent studies, presepsin has emerged as a promising early biomarker for various infections and a valuable tool in identifying sepsis and determining severity and prognosis [51].

Hypoalbuminemia has been identified as a predictive factor for IPN in a study [28]. Additionally, it is a recognized dose-dependent risk factor for organ failure, local complications, and malnutrition in acute pancreatitis [52].

Elevated serum LDH levels, indicating cellular injury, have demonstrated significant associations with severe acute pancreatitis and are recognized as a sensitive biomarker for pancreatic necrosis; nevertheless, their ability to predict IPN yielded an AUC of 0.77 [40,53]. However, when LDH levels were combined with lymphocyte levels in the late phase of AP, the AUC significantly increased to 0.94 [40].

HCT, BUN, and creatinine are indicators of organ perfusion and volume status and have been identified as predictors of the severity of AP and mortality [54]. HCT has limited predictive value for IPN, but creatinine and BUN have shown associations with IPN and have been incorporated into scoring systems [28,29].

Chen et al. examined the combined diagnostic accuracy of PCT, CRP, HCT, and BUN within the first 48 h of admission [29]. Furthermore, Wiese et al. developed a prediction model that included creatinine, CRP, albumin, and alcoholic etiology. These models demonstrated better ROC curves and AUC values compared to individual laboratory parameters alone [28]. After conducting a meta-analysis, it was found that patients with severe or necrotizing pancreatitis had a higher risk of developing IPN if they experienced over 50% necrosis of the pancreas, delayed enteral nutrition, and required invasive mechanical ventilation [55]. This highlights the potential benefit of a comprehensive approach in predicting infected necrosis [56,57].

### 4.1. Strengths and Limitations

Regarding the strengths of our study, we adhered to our pre-registered protocol, ensuring methodological rigor and transparency throughout the study. In contrast to the prior systematic review, we conducted a subgroup analysis based on the timing of the laboratory parameter measurements [48].

The main limitations of our study were the low number of eligible studies included and the observed variations in study design and patient populations. Additionally, the enrolled patients varied in terms of onset and severity of the disease, and this may impact the generalizability and applicability of the findings to a broader population. Furthermore, certain studies were of low quality. Another limitation is that due to the varying thresholds used in the studies and their unavailability in some cases, we cannot provide a recommended value for suspecting infection.

### 4.2. Implication for Practice and Research

The immediate application of scientific results is of great significance [56,57]. CRP and PCT as biomarkers may serve as valuable tools to facilitate appropriate antibiotic therapy in the late phase of AP. Future research with larger sample sizes and longer follow-up durations is needed to validate further and refine the use of CRP, PCT, creatinine, BUN, presepsin, and albumin in predicting IPN. Furthermore, developing a scoring system that combines multiple biomarkers and clinical parameters, such as imaging findings and patient demographics, may enhance the accuracy of predicting IPN.

## 5. Conclusions

The predictive value of CRP and PCT for infection is poor within 72 h after hospital admission. However, after the first three days of admission, the predictive value of CRP and PCT appears to be high enough for clinical use.

## Figures and Tables

**Figure 1 ijms-25-01273-f001:**
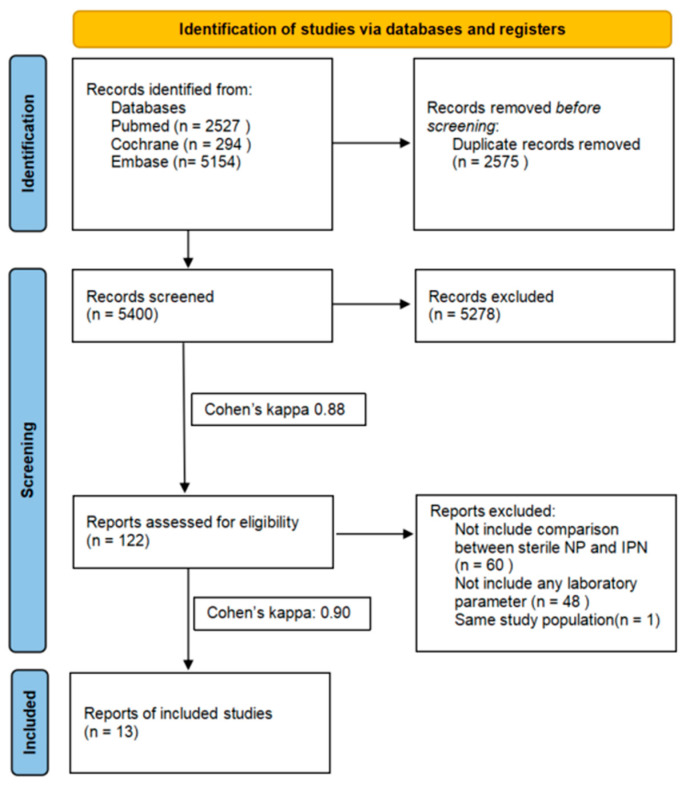
PRISMA 2020 flowchart representing the study selection process.

**Figure 2 ijms-25-01273-f002:**
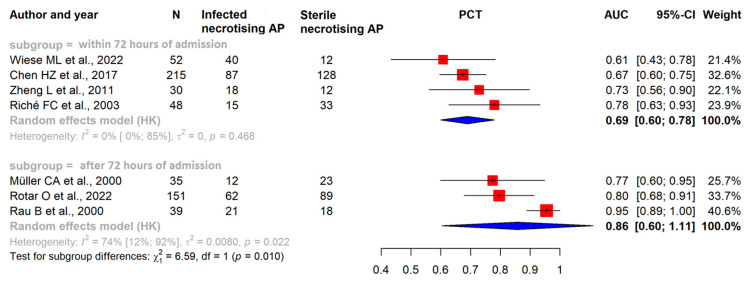
Forest plots representing the AUC of PCT within the initial 72 h of admission and 72 h after admission, examining a two-week period of the disease: AP: acute pancreatitis; AUC: area under the ROC curve; CI: confidence interval; h: hours; PCT: procalcitonin [28,29,30,31,32,33,34].

**Figure 3 ijms-25-01273-f003:**
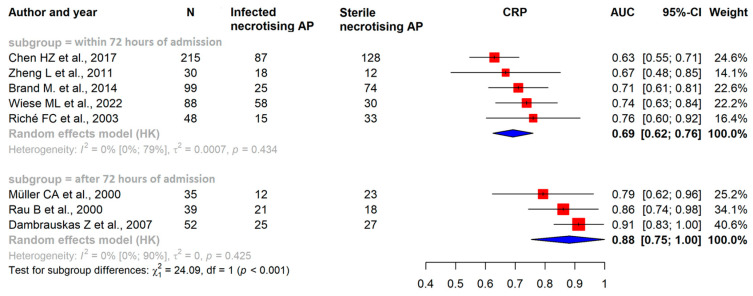
Forest plots representing the AUC of CRP within the initial 72 h of admission and 72 h after admission, examining a two-week period of the disease. AP: acute pancreatitis; AUC: area under the ROC curve; CI: confidence interval; CRP: C-reactive protein, h: hours [29,30,31,32,33,34,35].

**Figure 4 ijms-25-01273-f004:**
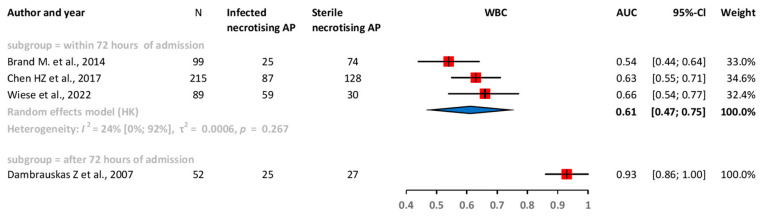
Forest plots representing the AUC of WBC within 72 h of admission. AP: acute pancreatitis; AUC: area under the ROC curve; CI: confidence interval; h: hours; WBC: white blood cell count [28,29,35,36].

**Table 1 ijms-25-01273-t001:** Basic characteristics of the included studies.

Publication Data	Study Design	Demography			Reference Test	Index Tests	Assessed Outcomes	Time Points for Laboratory Parameter Measurements	The Time Interval between Onset of Abdominal Pain and the Measurements of Laboratory Parameters
		Country	Population	Age (Years)					
**Block et al.** † (1987) [37]	cross-sectional (cohort-type accuracy study); prospective	Germany	161	N/A	surgery	Alb; Ca; HCT; WBC	Se, Sp	within 48 h	N/A
**Brand et al.** (2014) [36]	cross-sectional (cohort-type accuracy study); retrospective	Germany	99	52 ^a^ (18–84) ^b^	FNA	Alb, Ca, CRP, WBC	AUC; ROC; Se, Sp	within 36 h (most within 24 h)	N/A
**Chen et al.** (2017) [29]	cross-sectional (cohort-type accuracy study); retrospective	China	215	42.2 ^c^ (11.6) ^d^	CECT, US- or CT- guided FNA, invasive therapeutic procedures	BUN; Cr; CRP; D-dim; HCT; PCT; PLT; WBC	AUC; ROC; Se; Sp	within 48 h	<48 h before hospital admission
**Dambrauskas et al.** (2007) [35]	cross-sectional (cohort-type accuracy study); prospective	Lithuania	52	51.15 ^c^	CECT, FNA	CRP, WBC	AUC; NPV; PPV; ROC; Se; Sp	every fourth day until discharge	measurement of laboratory parameters occurred between days 21 and 40 after the onset of the disease in the subgroup analysis
**Mándi et al.** † (2000) [38]	cross-sectional (cohort-type accuracy study); prospective	Hungary	20	45.5 ^c^ (18.2) ^d^ (20–63) ^b^	CECT, US-guided FNA	IL-6; sICAM-1; PCT	NPV; PPV; Se; Sp	within 48 h, blood samples daily	N/A
**Müller et al.** (1999) [32]	cross-sectional (cohort-type accuracy study); prospective	Switzerland	35	56.3 ^c^ (27–87) ^b^	CECT, US- or CT-guided FNA	CRP; GCSF; PCT	AUC; ROC; Se; Sp	1–14 days daily and thereafter every third day	from day 0 until day 14 after the onset of the symptoms
**Rau et al.** (2000) [34]	cross-sectional (cohort-type accuracy study); prospective	Germany	61	(14–87) ^b^	CECT, US-guided FNA	CRP, PCT	AUC; ROC; Se; Sp	in 24 h intervals over 14 day	abdominal pain less than 120 h before hospital admission
**Riché et al.** (2003) [31]	cross-sectional (cohort-type accuracy study); prospective	France	48	(24–91) ^b^	CECT, CT-guided FNA, surgical drainage	CRP; IL-6; PCT; TNF-alpha	AUC, ROC	within 72 h daily	N/A
**Rotar et al.** (2022) [33]	cross-sectional (cohort-type accuracy study); prospective	Ukraine	151	(18–80) ^b^	CECT, therapeutic intervention,	PCT	AUC; ROC; Se; Sp	72 h before intervention	after the 4th weeks in case of 41 patients, before the 4th week in case of 74 patients
**Rotar et al.** † (2019) [39]	cross-sectional (cohort-type accuracy study); prospective	Ukraine	70	(18–80) ^b^	CECT, therapeutic intervention	Presepsin	AUC; ROC; Se; Sp	72 h	N/A
**Ueda et al.** † (2007) [40]	cross-sectional (cohort-type accuracy study); retrospective	Japan	75	52 ^c^ (2) ^d^	CECT, blood culture, US-guided FNA	LDH, Lymphocyte count	AUC; ROC	within 72 h	within 72 h
**Wiese et al.** (2022) [28]	cross-sectional (cohort-type accuracy study); retrospective	Germany	89	57.67 ^c^	CECT, PC drainage, EUS-guided FNA	Alb; BUN; Ca; Crea; CRP; HCT; IL-6; PCT	AUC; ROC; Se; Sp	within 48 h	N/A
**Zheng et al.** (2011) [30]	cross-sectional (cohort-type accuracy study); prospective	China	30	55.5 ^c^	CECT, US- or CT-guided FNA	CRP; IL-6, PCT; sTREM-1; TNF-alpha; WBC	AUC; NPV; PPV; ROC Se; Sp	72 h	N/A

a = median; b = range; c = mean; d = standard deviation; N/A = not applicable; Alb = albumin; AUC = area under the ROC curve; BUN = blood urea nitrogen; Ca = calcium; CECT = contrast-enhanced computed tomography; Crea = creatinine; CRP = c-reactive protein; D-dim = D-dimer; EUS = endoscopic ultrasound; FNA = fine needle aspiration; GCSF = granulocyte colony-stimulating factor; HCT = hematocrit; h = hours; IL-6 = interleukin-6; LDH = lactate dehydrogenase; NPV = negative predictive value; PCT = procalcitonin; PC = percutaneous; PLT = platelet; PPV = positive predictive value; ROC = receiver operating characteristic; Se = sensitivity; sICAM-1 = soluble intercelluar adhesion molecule-1; Sp = specificity; sTREM-1 = soluble triggering receptor expressed on myeloid cells; TNF-alpha = tumor necrosis factor-alpha; US = ultrasound; WBC = white blood cell; † = study included only in a systematic review.

## Data Availability

The datasets used in this study can be found in the full-text articles included in the systematic review and meta-analysis.

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
