# Peer review of "Persistently High Procalcitonin and C-Reactive Protein Are Good Predictors of Infection in Acute Necrotizing Pancreatitis: A Systematic Review and Meta-Analysis"

_ijms, 2024, doi:10.3390/ijms25021273_

Round 1

Reviewer 1 Report

Comments and Suggestions for Authors

I've carefully read the systematic review and meta-analysis on the role of biomarkers - CRP and procalcitonin, as predictors of infection in the late phase of acute pancreatitis - which is an interesting paper summarising current literature data on the topic, with meaningful impact in clinical practice considering the excess of antibiotic prescriptions in AP.

Methodology is robust and results are very well represented.

The title should be more precise, as the SR and MA is referring to infection in acute necrotic collections in AP cases and not infection in general (some may be biliary AP with cholangitis or other infections).

I see a limitation in defining the early phase of AP for the first 72 hours only, followed by a late phase, as the inflammatory phase usually lasts more and infectious complications develop after the first week. 

Author Response

Dear Reviewer,

Thank you for your review. We greatly appreciate your comments, which have significantly improved our manuscript. Below, you will find our point-by-point response to your feedback. We hope that our revised manuscript meets your expectations and is suitable for publication. Please let us know if any further changes are required.

  • Comment 1: I've carefully read the systematic review and meta-analysis on the role of biomarkers - CRP and procalcitonin, as predictors of infection in the late phase of acute pancreatitis - which is an interesting paper summarising current literature data on the topic, with meaningful impact in clinical practice considering the excess of antibiotic prescriptions in AP.

          Methodology is robust and results are very well represented.

The title should be more precise, as the SR and MA is referring to infection in acute necrotic collections in AP cases and not infection in general (some may be biliary AP with cholangitis or other infections).

Response 1: We appreciate your positive feedback on the manuscript! We agree with you and adjusted the title of the manuscript to be more precise.

Action 1: The title has been revised; we changed it to: Persistently High Procalcitonin and C-Reactive Protein are Good Predictors of Infection in Acute Necrotizing Pancreatitis.

  • Comment 2: I see a limitation in defining the early phase of AP for the first 72 hours only, followed by a late phase, as the inflammatory phase usually lasts more and infectious complications develop after the first week.

Response 2: Thank you for the guidance on this matter. We agree with this comment.

Action 2: We have made some changes to the terminology used for the management of acute pancreatitis. Specifically, we have replaced the phrases "early and late phases" with "within 72 hours of admission" and "after 72 hours of admission" respectively to improve clarity. Additionally, the studies focusing on the late phase of the disease have covered a period of at least two weeks.

Reviewer 2 Report

Comments and Suggestions for Authors

The manuscript “Persistently High Procalcitonin and C-Reactive Protein are 2 Good Predictors of Infection in the Late Phase of Acute Pancreatitis: A Systematic Review and Meta-Analysis” by Tarján et al. highlights important aspects of treatment of infected necrotizing pancreatitis (INP) that is correlated with a high risk of mortality. With the lack of specific laboratory markers, it is very difficult to identify the timeline of the application of antibiotic therapy for these patients. Authors have provided compelling evidence based on systematic review and meta-analysis, with detailed methodology appropriately reported, that C-reactive protein and procalcitonin increased levels can be used as a prediction tool of INP at the late phase of acute pancreatitis with a recommendation to initiate relevant antibiotic therapy.

The manuscript is nicely written and presented with all the necessary evidence provided to support the conclusions.

Author Response

Dear Reviewer,

Thank you for your review. We greatly appreciate your comment.

I also would like to take the opportunity to wish you a Happy New Year!

Best regards,

Péter Hegyi

  • Comment The manuscript “Persistently High Procalcitonin and C-Reactive Protein are 2 Good Predictors of Infection in the Late Phase of Acute Pancreatitis: A Systematic Review and Meta-Analysis” by Tarján et al. highlights important aspects of treatment of infected necrotizing pancreatitis (INP) that is correlated with a high risk of mortality. With the lack of specific laboratory markers, it is very difficult to identify the timeline of the application of antibiotic therapy for these patients. Authors have provided compelling evidence based on systematic review and meta-analysis, with detailed methodology appropriately reported, that C-reactive protein and procalcitonin increased levels can be used as a prediction tool of INP at the late phase of acute pancreatitis with a recommendation to initiate relevant antibiotic therapy. The manuscript is nicely written and presented with all the necessary evidence provided to support the conclusions.
  • Response: We appreciate your positive feedback on the manuscript!

Reviewer 3 Report

Comments and Suggestions for Authors

Manuscript ID: ijms-2742568

Title: Persistently High Procalcitonin and C-Reactive Protein are Good Predictors of Infection in the Late Phase of Acute Pancreatitis: A Systematic Review and Meta-Analysis

Authors: Tarjan et al.

The manuscript is unclear and misleading. The manuscript should be in line with the current recommendations in the diagnosis and therapy of infected pancreatic necrosis. Inconsistence between the current guidelines and the concepts presented in the manuscript could eventually be acceptable if the manuscript presented well-documented observation that the current recommendations are incorrect. The abstract does not contain detailed information about what was assessed, the final conclusions is questionable. Therefore, in the reviewer opinion, the manuscript should be rejected.

List of the most important imperfections and errors:

  1. The authors should know and present the most important, current recommendations of American Gastroenterological Association regarding prevention and treatment of infected pancreatic necrosis (PMID: 31479658). In particular, firstly they should write that antimicrobial therapy is best indicated for culture-proven infection in pancreatic necrosis or when infection is strongly suspected (gas in the collection, bacteremia, sepsis, or clinical deterioration). Routine use of prophylactic antibiotics to prevent infection of sterile necrosis is not recommended. Secondly, when infected necrosis is suspected, broad-spectrum intravenous antibiotics with ability to penetrate pancreatic necrosis should be favored (carbapenems, quinolones, and metronidazole). Routine use of antifungal agents is not recommended. Computed tomography-guided fine-needle aspiration for Gram stain and cultures is unnecessary in the majority of cases. Thirdly, in patients with pancreatic necrosis, enteral feeding should be initiated early to decrease the risk of infected necrosis. A trial of oral nutrition is recommended immediately in patients in whom there is absence of nausea and vomiting and no signs of severe ileus or gastrointestinal luminal obstruction. When oral nutrition is not feasible, enteral nutrition by either nasogastric/duodenal or nasojejunal tube should be initiated as soon as possible. Total parenteral nutrition should be considered only in cases where oral or enteral feeds are not feasible or tolerated. Fourthly, drainage and/or debridement of pancreatic necrosis is indicated in patients with infected necrosis. Drainage and/or debridement may be required in patients with sterile pancreatic necrosis and persistent unwellness marked by abdominal pain, nausea, vomiting, and nutritional failure or with associated complications, including gastrointestinal luminal obstruction; biliary obstruction; recurrent acute pancreatitis; fistulas; or persistent systemic inflammatory response syndrome. Fifthly, pancreatic debridement should be avoided in the early, acute period (first 2 weeks), as it has been associated with increased morbidity and mortality. Debridement should be optimally delayed for 4 weeks and performed earlier only when there is an organized collection and a strong indication. Sixthly, percutaneous drainage and transmural endoscopic drainage are both appropriate first-line, nonsurgical approaches in managing patients with walled-off pancreatic necrosis (WON). Endoscopic therapy through transmural drainage of WON may be preferred, as it avoids the risk of forming a pancreatocutaneous fistula.
  2. Abstract, lines 33-37. The authors stated, “After the first 72 hours, in the late phase (probably acute pancreatitis?)”. This statement is unclear. According to the revised Atlanta classification (PMID: 23100216), early phase of acute is usually over by the end of the first week but may extend into the second week During this phase systemic disturbances result from the host response to local pancreatic injury. Cytokine cascades are activated by the pancreatic inflammation which manifest clinically as the systemic inflammatory response syndrome (SIRS). For this reason, the authors should define exactly what “After the first 72 hours, in the late phase” means. Does this mean the time point after 72 h from the patient’s admission to the hospital? If so, it will not be a late phase of the disease, but still early. The authors should specify exactly how long after admission, the tests were performed. This information must also be presented in the Methods, Results, Figures and Discussion.
  3. Abstract, lines 34-37. The authors present the area under the ROC curve (AUC) for C-reactive protein, procalcitonin and for white blood cell count. However, it is not known how the presented AUC values were obtained. What patients’ group was used for comparison when determining AUC in patients with infected necrotizing pancreatitis (INP)? This information must also be presented in the Methods, Results, Figures and Discussion.
  4. Abstract, Results, Figures. Based on the articles used, the authors should present the values of C-reactive protein (CRP) and procalcitonin, above which antibiotic therapy would be recommended.
  5. Abstract, line 37-40. In conclusions, the authors stated that ”The predictive value of CRP and PCT for infection is poor in the early phase (of acute pancreatitis?) but seems good in the late phase of the disease. Based on these results, infection is likely in case of persistently high CRP and PCT, and antibiotic initiation may be recommended” And the conclusions at the end of the manuscript (lines 327-329) “The predictive value of CRP and PCT for infection is poor in the early phase of AP. However, in the late phase of the disease, the predictive value of CRP and PCT appears to be high enough for clinical use.” What is prognostic significance of CRP and PCT values in the treatment of infected necrotizing pancreatitis? Do they have any significance, or do they just seem to be good and may have unspecified value in clinical settings?

Author Response

Dear Reviewer,

Thank you for your review. We greatly appreciate your comments, which have significantly improved our manuscript. Below, you will find our point-by-point response to your feedback. We hope that our revised manuscript meets your expectations and is suitable for publication. Please let us know if any further changes are required.

I also would like to take the opportunity to wish you a Happy New Year!

Best regards,

Péter Hegyi

The manuscript is unclear and misleading. The manuscript should be in line with the current recommendations in the diagnosis and therapy of infected pancreatic necrosis. Inconsistence between the current guidelines and the concepts presented in the manuscript could eventually be acceptable if the manuscript presented well-documented observation that the current recommendations are incorrect. The abstract does not contain detailed information about what was assessed, the final conclusions is questionable. Therefore, in the reviewer opinion, the manuscript should be rejected.

List of the most important imperfections and errors:

Comment 1: The authors should know and present the most important, current recommendations of American Gastroenterological Association regarding prevention and treatment of infected pancreatic necrosis (PMID: 31479658). In particular, firstly they should write that antimicrobial therapy is best indicated for culture-proven infection in pancreatic necrosis or when infection is strongly suspected (gas in the collection, bacteremia, sepsis, or clinical deterioration). Routine use of prophylactic antibiotics to prevent infection of sterile necrosis is not recommended. Secondly, when infected necrosis is suspected, broad-spectrum intravenous antibiotics with ability to penetrate pancreatic necrosis should be favored (carbapenems, quinolones, and metronidazole). Routine use of antifungal agents is not recommended. Computed tomography-guided fine-needle aspiration for Gram stain and cultures is unnecessary in the majority of cases. Thirdly, in patients with pancreatic necrosis, enteral feeding should be initiated early to decrease the risk of infected necrosis. A trial of oral nutrition is recommended immediately in patients in whom there is absence of nausea and vomiting and no signs of severe ileus or gastrointestinal luminal obstruction. When oral nutrition is not feasible, enteral nutrition by either nasogastric/duodenal or nasojejunal tube should be initiated as soon as possible. Total parenteral nutrition should be considered only in cases where oral or enteral feeds are not feasible or tolerated. Fourthly, drainage and/or debridement of pancreatic necrosis is indicated in patients with infected necrosis. Drainage and/or debridement may be required in patients with sterile pancreatic necrosis and persistent unwellness marked by abdominal pain, nausea, vomiting, and nutritional failure or with associated complications, including gastrointestinal luminal obstruction; biliary obstruction; recurrent acute pancreatitis; fistulas; or persistent systemic inflammatory response syndrome. Fifthly, pancreatic debridement should be avoided in the early, acute period (first 2 weeks), as it has been associated with increased morbidity and mortality. Debridement should be optimally delayed for 4 weeks and performed earlier only when there is an organized collection and a strong indication. Sixthly, percutaneous drainage and transmural endoscopic drainage are both appropriate first-line, nonsurgical approaches in managing patients with walled-off pancreatic necrosis (WON). Endoscopic therapy through transmural drainage of WON may be preferred, as it avoids the risk of forming a pancreatocutaneous fistula.

 Response 1: Thank you for pointing this out.

Action 1: Considering your recommendation, we have integrated this aspect into the Introduction section.

Page 2, Line 63-73: “According to the American College of Gastroenterology Guideline, patients demonstrating worsening or a lack of improvement after 7–10 days of hospitalization with pancreatic or extrapancreatic necrosis should be evaluated for the potential presence of infected necrosis. For such patients, the recommendation includes either (i) conducting an initial computed tomography-guided fine needle aspiration (CT FNA) for Gram stain and culture to inform the appropriate use of antibiotics, or (ii) initiating empiric antibiotic treatment without CT FNA. In cases of confirmed infected necrosis, the use of antibiotics that can penetrate pancreatic necrosis may be beneficial in delaying intervention, thereby reducing morbidity and mortality. In stable patients diagnosed with infected necrosis, the initiation of surgical, radiologic, and/or endoscopic drainage is recommended to be deferred, preferably for 4 weeks [10].”

Comment 2: Abstract, lines 33-37. The authors stated, “After the first 72 hours, in the late phase (probably acute pancreatitis?)”. This statement is unclear. According to the revised Atlanta classification (PMID: 23100216), early phase of acute is usually over by the end of the first week but may extend into the second week During this phase systemic disturbances result from the host response to local pancreatic injury. Cytokine cascades are activated by the pancreatic inflammation which manifest clinically as the systemic inflammatory response syndrome (SIRS). For this reason, the authors should define exactly what “After the first 72 hours, in the late phase” means. Does this mean the time point after 72 h from the patient’s admission to the hospital? If so, it will not be a late phase of the disease, but still early. The authors should specify exactly how long after admission, the tests were performed. This information must also be presented in the Methods, Results, Figures and Discussion.

Response 2: Thank you for the guidance on this matter. We agree with this comment.

Action 2: We have made some changes to the terminology used for the management of acute pancreatitis. Specifically, we have replaced the phrases "early and late phases" with "within 72 hours of admission" and "after 72 hours of admission" respectively to improve clarity. Additionally, the studies focusing on the late phase of the disease have covered a period of at least two weeks.

Comment 3: Abstract, lines 34-37. The authors present the area under the ROC curve (AUC) for C-reactive protein, procalcitonin and for white blood cell count. However, it is not known how the presented AUC values were obtained. What patients’ group was used for comparison when determining AUC in patients with infected necrotizing pancreatitis (INP)? This information must also be presented in the Methods, Results, Figures and Discussion.

Response 3: Thank you for your comment. We compared the laboratory data between sterile necrotizing pancreatitis and INP.

Action 3: We augmented the Abstract, Methods, and Discussion section of our manuscript to provide a clearer definition of the control group. Page 1, line 33 “Laboratory biomarkers compared between sterile necrotizing pancreatitis and INP were used as the index test, and our outcome measures included sensitivity, specificity, the receiver operating characteristic (ROC) curve and area under the ROC curve (AUC).” Page 3, line 107 “(3) the index test was using any laboratory biomarker that were compared between patients suffering from sterile necrotizing pancreatitis (SNP)or INP;” Page 9, line 255 “Our meta-analysis found that the effectiveness of CRP and PCT in predicting infection in necrotizing pancreatitis”

Comment 4: Abstract, Results, Figures. Based on the articles used, the authors should present the values of C-reactive protein (CRP) and procalcitonin, above which antibiotic therapy would be recommended.

Response 4: Thank you for pointing this out, unfortunately, we could not provide a recommendation for the threshold due to the variation in thresholds used in the studies and their limited availability. Further clinical trials are needed to clarify this question

Action 4: We have augmented the Limitations section of our manuscript to provide a more comprehensive and nuanced discussion of this aspect. Page 11, Line 344 “Another limitation is that due to the varying thresholds used in the studies and their unavailability in some cases, we cannot provide a recommended value for suspecting infection.”

Comment 5: Abstract, line 37-40. In conclusions, the authors stated that ”The predictive value of CRP and PCT for infection is poor in the early phase (of acute pancreatitis?) but seems good in the late phase of the disease. Based on these results, infection is likely in case of persistently high CRP and PCT, and antibiotic initiation may be recommended” And the conclusions at the end of the manuscript (lines 327-329) “The predictive value of CRP and PCT for infection is poor in the early phase of AP. However, in the late phase of the disease, the predictive value of CRP and PCT appears to be high enough for clinical use.” What is prognostic significance of CRP and PCT values in the treatment of infected necrotizing pancreatitis? Do they have any significance, or do they just seem to be good and may have unspecified value in clinical settings?

Response 5:

Thank you for your comment.

The prognostic significance of C-reactive protein (CRP) and procalcitonin (PCT) values in the treatment of infected necrotizing pancreatitis (INP) is notable, but their utility appears to vary depending on the phase of acute pancreatitis (AP).

Within 72 hours after hospital admission, CRP and PCT may have a limited predictive value for detecting infection. Their poor performance during this period suggests caution in relying solely on these biomarkers for early infection detection (PMID: 31551798). After the first three days of admission, investigating at least two weeks of the period of the disease, both CRP and PCT demonstrate improved predictive value for infection. Elevated levels indicate a higher likelihood of infection, suggesting their potential clinical utility for treatment decisions. Persistently high CRP and PCT levels after the first three days of admission, investigating at least two weeks of the period of the disease, may signal ongoing infection, supporting the recommendation for antibiotic initiation. CRP and PCT, being dynamic markers of inflammation and infection, play a crucial role in monitoring disease progression and treatment response. While they may not be definitive standalone indicators, their trends and patterns aid clinicians in making informed decisions about treatment strategies, especially in the late phase.

In conclusion, CRP and PCT values hold significant prognostic value, particularly in the later stages of AP and INP. Their trends can serve as valuable indicators for ongoing infection and may guide clinical decisions, including the initiation of antibiotic therapy. However, their predictive value may be limited in within the first three days, highlighting the need for a comprehensive clinical assessment and potentially the integration of additional diagnostic modalities for accurate and timely decision-making.

Action 5: No action.

Round 2

Reviewer 3 Report

Comments and Suggestions for Authors

Manuscript ID: ijms-2742568

Title: Persistently High Procalcitonin and C-Reactive Protein are Good Predictors of Infection in the Late Phase of Acute Pancreatitis: A Systematic Review and Meta-Analysis

Authors: Tarjan et al. 

The new version of the manuscript shows some improvement, but there are still numerous inaccuracies and omissions that require correction.

  1. Abstract and other parts of the manuscript. Previous comment 2. The authors have stated that “we have replaced the phrases "early and late phases" with "within 72 hours of admission" and "after 72 hours of admission" respectively to improve clarity. Additionally, the studies focusing on the late phase of the disease have covered a period of at least two weeks.” In addition, they have stated that “The predictive value of CRP and PCT for infection is poor within 72 hours after hospital admission but seems good after the first 72 h”. What does it mean? This statement suggests that already on the 4th or 5th day after admission to the hospital the measurement of procalcitonin and C-Reactive Protein has a prognostic significance in determining the risk of developing infected necrotizing pancreatitis. Are they sure about this? The development of pancreatic necrosis needs several days, and infected necrosis is rare during the first week. (PMID: 23100216). The authors should better define the time frame when the prognostic significance of procalcitonin and C-RP in detecting the risk of developing infected necrotizing pancreatitis. This should be presented in the abstract, results and discussion.
  2. Introduction, lines 52-53. The authors stated that “While interstitial edematous pancreatitis is the most common form, pancreatic necrosis occurs in approximately 20% of cases [2]”. This statement is supported by article “Endoscopic Management of Infected Necrotizing Pancreatitis: an Evidence-Based Approach.” Indeed, in the mentioned article, their authors stated that “Approximately 20% of patients develop a severe pancreatitis with necrosis of the (peri)pancreatic tissue. However, the references shown in their statement (PMID: 17032204) does not supports this claim. Therefore, it would be advisable for authors of the manuscript to present real data in this regard. According to Banks et al. (PMID: 23100216). “About 5–10% of patients develop necrosis of the pancreatic parenchyma, the peripancreatic tissue or both” in the course of acute pancreatitis.
  3. Introduction, previous comment 1. For unknown reasons, instead of the suggested article (PMID: 31479658) from 2020 regarding the American Gastroenterological Association’s guidelines for pancreatic necrosis in acute pancreatitis, the authors used and article from 2013 regarding not pancreatic necrosis in pancreatitis, but acute pancreatitis in general. The reviewer recommends using the early suggested article, which would avoid sentence such as “However, definitive confirmation necessitates microbiological testing following fine-needle aspiration or drainage procedures” (lines 65-66). Additionally, the authors, in line with previous comment 1, should carefully present the AGA guideline shown in suggested article, especially regarding preventive measures, such as early enteral nutrition, which increases the integrity of the mucosal barrier and reduces the possibility of bacterial translocation in the gastrointestinal tract (PMID: 37111158).
  4. Introduction. Pancreatic necrosis is a result of the impairment of pancreatic perfusion (PMID: 23100216). In addition, clinical (PMID: 8831599; PMID: 10669996; PMID: 686887) and experimental studies (PMID: 11453102; PMID: 2252994) indicate that the pancreatic ischemia/reperfusion injury plays an essential role in the development of acute pancreatitis. Pancreatic ischemia/reperfusion injury may be a primary cause of acute pancreatitis, but also in acute pancreatitis caused by other, primary non-vascular mechanisms, the early disturbance of pancreatic circulation is observed and is associated with the progression of the severity of acute pancreatitis. On the other hand, the improvement in blood flow prevents the development of acute pancreatitis and accelerates recovery in this disease (PMID: 18955758; PMID: 37371528; PMID: 37865961). The authors should write some words on this topic, as well as that aggressive hydration should be provided to all patients, unless cardiovascular and/or renal comorbidities preclude it, and early aggressive intravenous hydration is most beneficial within the first 12-24 h (PMID: 23896955).
  5. Table 1. The table should have a heading explaining what it concerns. The current form “Table 1. This is a table. Tables should be placed in the main text near to the first time they are cited” is not acceptable. In addition, what does “time points” mean? There is no data whether the presented values refer to the first hospitalization, the second hospitalization in a secondary hospital or the third hospitalization in a tertiary hospital. Do the authors have such data? In accordance with recommendations of “Classification of acute pancreatitis—2012: revision of the Atlanta classification” (PMID: 23100216), “The onset of acute pancreatitis is defined as the time of onset of abdominal pain (not the time of admission to the hospital). The time interval between onset of abdominal pain and first admission to the hospital should be noted. When patients with severe disease are transferred to a tertiary hospital, the intervals between onset of symptoms, first admission and transfer should be noted. Data recorded from a tertiary care hospital should be stratified to allow separate consideration of the outcomes of patients who were admitted directly and those admitted by transfer from another hospital.”

Author Response

Dear Reviewer,

Thank you for your review. We greatly appreciate your comments, which have significantly improved our manuscript. Below, you will find our point-by-point response to your feedback. We hope that our revised manuscript meets your expectations and is suitable for publication. Please let us know if any further changes are required.

Best regards,

Péter Hegyi

The new version of the manuscript shows some improvement, but there are still numerous inaccuracies and omissions that require correction.

Comment 1: Abstract and other parts of the manuscript. Previous comment 2. The authors have stated that “we have replaced the phrases "early and late phases" with "within 72 hours of admission" and "after 72 hours of admission" respectively to improve clarity. Additionally, the studies focusing on the late phase of the disease have covered a period of at least two weeks.” In addition, they have stated that “The predictive value of CRP and PCT for infection is poor within 72 hours after hospital admission but seems good after the first 72 h”. What does it mean? This statement suggests that already on the 4th or 5th day after admission to the hospital the measurement of procalcitonin and C-Reactive Protein has a prognostic significance in determining the risk of developing infected necrotizing pancreatitis. Are they sure about this? The development of pancreatic necrosis needs several days, and infected necrosis is rare during the first week. (PMID: 23100216). The authors should better define the time frame when the prognostic significance of procalcitonin and C-RP in detecting the risk of developing infected necrotizing pancreatitis. This should be presented in the abstract, results and discussion.

Response 1: Thank you for your insightful comment. In the context of our meta-analysis, where we extensively relied on data available in the literature, certain limitations and uncertainties arise. For instance, some articles did not explicitly specify the exact time points for the utilized laboratory parameters (as observed in Dambrauskas, Müller, Rau). Given the diverse course of acute pancreatitis, it is probable that the laboratory results from the 4-5 day period may not have been exclusively attributed to that timeframe, but this was not definitively indicated. Consequently, the risk of bias assessment received an "unclear" rating in the timing domain, contributing to a low precision evaluation in the GRADE assessment.

In light of these limitations, the literature did not provide a more precise timeframe. However, in the early phase, we could confidently assert the values within the first 72 hours.

It is important to note that, based on the available literature, we couldn't definitively pinpoint the day from which the prognostic effect of CRP and PCT becomes more pronounced. Nevertheless, our analysis consistently indicated a stronger effect after 72 hours compared to within the initial 72 hours. While the raw data from various centers could potentially allow for subgrouping based on the day of the examination of laboratory biomarkers, this would necessitate a different study design and hypothesis.

We appreciate your thorough consideration of these complexities and remain committed to providing a transparent and comprehensive account of our findings.

Action 1:

Page 9 line 230: “Forest plots representing the AUC of PCT within the initial 72 hours of admission and 72 hours after admission, examining a two-week period of the disease.”

Page 9 line 234: “Forest plots representing the AUC of CRP within the initial 72 hours of admission and 72 hours after admission, examining a two-week period of the disease.” 

Page 10 line 243: “After the first 72 hours of admission, in studies investigating a minimum two-week period of the disease, the pooled AUC of CRP showed an elevated level of 0.88 (CI: 0.75-1.00); for PCT, it was 0.86 (CI: 0.60-1.11), which shows that it had a good predictive value.”

Comment 2: Introduction, lines 52-53. The authors stated that “While interstitial edematous pancreatitis is the most common form, pancreatic necrosis occurs in approximately 20% of cases [2]”. This statement is supported by article “Endoscopic Management of Infected Necrotizing Pancreatitis: an Evidence-Based Approach.” Indeed, in the mentioned article, their authors stated that “Approximately 20% of patients develop a severe pancreatitis with necrosis of the (peri)pancreatic tissue. However, the references shown in their statement (PMID: 17032204) does not supports this claim. Therefore, it would be advisable for authors of the manuscript to present real data in this regard. According to Banks et al. (PMID: 23100216). “About 5–10% of patients develop necrosis of the pancreatic parenchyma, the peripancreatic tissue or both” in the course of acute pancreatitis.

Response 2: We appreciate your thorough review of our manuscript. Upon your suggestion, we have revisited the literature, including the reference by Banks et al. (PMID: 23100216), and indeed, it provides a more accurate representation of the incidence. We will amend the introduction accordingly to reflect that about 5–10% of patients develop necrosis.

Action 2: Page 2 line 48 “While interstitial edematous pancreatitis is the most common form, pancreatic necrosis occurs in approximately 5-10% of cases.”

Comment 3: Introduction, previous comment 1. For unknown reasons, instead of the suggested article (PMID: 31479658) from 2020 regarding the American Gastroenterological Association’s guidelines for pancreatic necrosis in acute pancreatitis, the authors used and article from 2013 regarding not pancreatic necrosis in pancreatitis, but acute pancreatitis in general. The reviewer recommends using the early suggested article, which would avoid sentence such as “However, definitive confirmation necessitates microbiological testing following fine-needle aspiration or drainage procedures” (lines 65-66). Additionally, the authors, in line with previous comment 1, should carefully present the AGA guideline shown in suggested article, especially regarding preventive measures, such as early enteral nutrition, which increases the integrity of the mucosal barrier and reduces the possibility of bacterial translocation in the gastrointestinal tract (PMID: 37111158).

Response 3: Thank you for your thoughtful review and constructive suggestions. We agree that using the updated article will enhance the accuracy and relevance of our introduction.

Action 3: Page 2 line 70 “Antimicrobial therapy is most appropriate when there is a culture-proven infection in pancreatic necrosis or a strong suspicion of infection indicated by factors such as gas in the collection, bacteremia, sepsis, or clinical deterioration. Prophylactic antibiotics for preventing infection of sterile necrosis are not recommended. Initiating enteral feeding early in patients with pancreatic necrosis is recommended to reduce the risk of infection. This preventive measure enhances the integrity of the mucosal barrier and reduces the likelihood of bacterial translocation in the gastrointestinal tract. Drainage and/or debridement of pancreatic necrosis is warranted in patients with infected necrosis. It is recommended to avoid pancreatic debridement in the early acute period (first 2 weeks) due to its association with increased morbidity and mortality, optimal debridement is ideally delayed for 4 weeks.”

Comment 4: Introduction. Pancreatic necrosis is a result of the impairment of pancreatic perfusion (PMID: 23100216). In addition, clinical (PMID: 8831599; PMID: 10669996; PMID: 686887) and experimental studies (PMID: 11453102; PMID: 2252994) indicate that the pancreatic ischemia/reperfusion injury plays an essential role in the development of acute pancreatitis. Pancreatic ischemia/reperfusion injury may be a primary cause of acute pancreatitis, but also in acute pancreatitis caused by other, primary non-vascular mechanisms, the early disturbance of pancreatic circulation is observed and is associated with the progression of the severity of acute pancreatitis. On the other hand, the improvement in blood flow prevents the development of acute pancreatitis and accelerates recovery in this disease (PMID: 18955758; PMID: 37371528; PMID: 37865961). The authors should write some words on this topic, as well as that aggressive hydration should be provided to all patients, unless cardiovascular and/or renal comorbidities preclude it, and early aggressive intravenous hydration is most beneficial within the first 12-24 h (PMID: 23896955).

Response 4: Thank you for your comment. We have expanded the introduction on pancreatic ischemia/reperfusion injury, its role in the development of acute pancreatitis.

Action 4: Page 2 line 52 “Pancreatic necrosis arises from compromised pancreatic perfusion, emphasizing the pivotal role of pancreatic ischemia and microcirculatory disturbances in acute pancreatitis development. This injury, whether primary or secondary to non-vascular causes, manifests early in acute pancreatitis and correlates with severity progression. Conversely, enhancing blood flow not only averts acute pancreatitis but also expedites recovery. Therefore, it is recommended to provide aggressive hydration to all patients, especially during the first 12-24 hours, unless restricted by cardiovascular or renal comorbidities.”

Comment 5: Table 1. The table should have a heading explaining what it concerns. The current form “Table 1. This is a table. Tables should be placed in the main text near to the first time they are cited” is not acceptable. In addition, what does “time points” mean? There is no data whether the presented values refer to the first hospitalization, the second hospitalization in a secondary hospital or the third hospitalization in a tertiary hospital. Do the authors have such data? In accordance with recommendations of “Classification of acute pancreatitis—2012: revision of the Atlanta classification” (PMID: 23100216), “The onset of acute pancreatitis is defined as the time of onset of abdominal pain (not the time of admission to the hospital). The time interval between onset of abdominal pain and first admission to the hospital should be noted. When patients with severe disease are transferred to a tertiary hospital, the intervals between onset of symptoms, first admission and transfer should be noted. Data recorded from a tertiary care hospital should be stratified to allow separate consideration of the outcomes of patients who were admitted directly and those admitted by transfer from another hospital.”

Response 5: Thank you for pointing this out, in one article (Wiese ML. et al), there was information that patients were transferred; however, in this hospitalization period, the duration of the previous hospitalization was also included. The other articles did not mention transfers or admitted patients from other healthcare facilities.

Action 5:

We have augmented the table (Table 1. Basic characteristics of the included studies.) by adding a new column that precisely indicates the duration between the onset of symptoms and the first admission. This addition aims to offer a clearer understanding of the time intervals involved.

Round 3

Reviewer 3 Report

Comments and Suggestions for Authors

The current version of the manuscript is suitable for publication.